# EXPLORING THE DESIGN SPACE OF AUTOREGRESSIVE MODELS FOR EFFICIENT AND SCALABLE IMAGE GENERATION

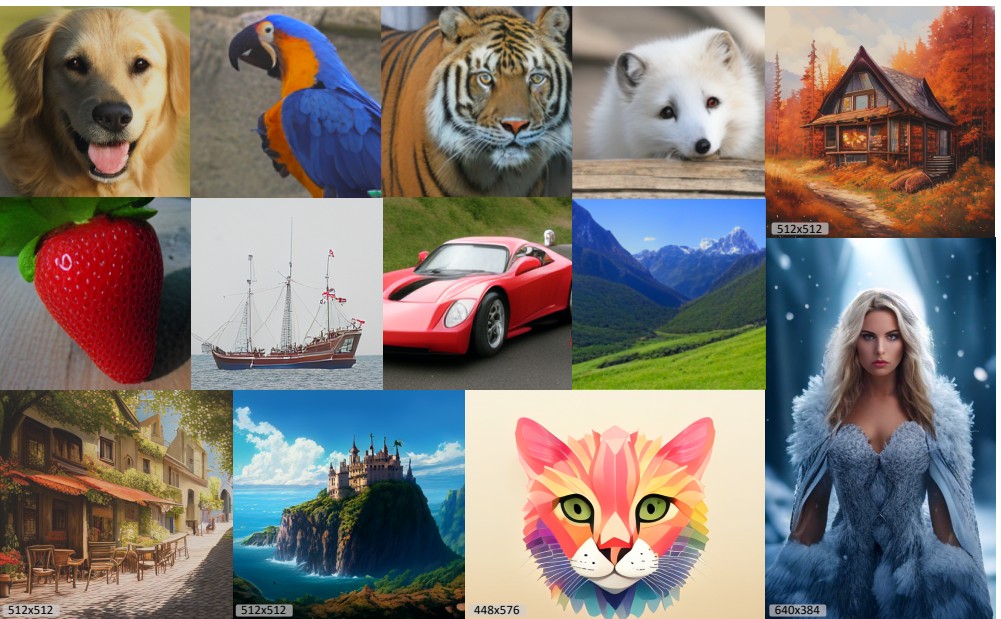

Figure 1: **Image generation with our MaskGIL models.** We show samples from our class-conditional generation (top left) and text-conditional generation in various resolutions.

## ABSTRACT

Autoregressive (AR) models and their variants are re-revolutionizing visual generation with improved frameworks. However, unlike the well-established practices for diffusion models, a comprehensive recipe for AR models is lacking, *e.g.*, selecting image tokenizers, model architectures, and AR paradigms. In this work, we delve into the design space of general AR models, including Mask Autoregressive (MAR) models, to identify optimal configurations for efficient and scalable image generation. We first conduct a detailed evaluation of four prevalent image tokenizers across both AR and MAR settings, examining the impact of codebook size (ranging from 1,024 to 262,144) on generation quality, and identify the most effective tokenizer for image generation. Building on these insights, we propose an enhanced MAR model architecture, named **Mask**ed **G**enerative **I**mage **L**LaMA (**MaskGIL**), comprising of LlamaGen-VQ and Bidirectional LLaMA. To ensure stable scaling, we introduce modifications like query-key normalization and post-normalization, resulting in a series of class-conditional MaskGIL models, ranging from 111M to 1.4B parameters. MaskGIL significantly improves the MAR baseline, achieving a 3.71 FID comparable to state-of-the-art AR models on the ImageNet $256{\times}256$ benchmark, with only 8 inference steps, far fewer than the 256 steps needed for AR models. We also introduce a text-conditional MaskGIL model with 775M parameters, capable of flexibly generating images at any resolution. To bridge AR and MAR image generation, we explore their combination during inference time. We release all models and code to foster further research[1].

---

[1]https://anonymous.4open.science/r/ICLR-1299

# 1 INTRODUCTION

Autoregressive (AR) generative models have garnered increasing attention in image generation (Razavi et al., 2019; Esser et al., 2021; Ramesh et al., 2021b; Lee et al., 2022; Yu et al., 2022; Sun et al., 2024; Liu et al., 2024), inspired by the success of Transformer (Vaswani, 2017) and GPT (Brown, 2020; OpenAI, 2022; Achiam et al., 2023) in NLP. This paradigm typically unfolds in two stages: the first stage is to quantize an image to a sequence of discrete tokens. In the second stage, an AR model is trained to predict the next token sequentially, based on the previously generated tokens. While AR models exhibit strong generative capabilities, their efficiency is hindered by the extensive number of inference steps, as the AR model *generates one token at a time*.

To overcome this limitation, Mask Autoregressive (MAR) generative models (Chang et al., 2022; Lezama et al., 2022; Li et al., 2023; Qian et al., 2023; Chang et al., 2023; Li et al., 2024b) have been developed, aiming to deliver high-quality image generation with fewer inference steps. Unlike traditional AR models that predict the next token, MAR models predict *a subset of tokens*, offering a potential speed advantage. However, due to the inherent difficulties in training and prediction, the generative capabilities of existing MAR models remain less robust compared to AR models.

In addition, the choice of discrete-value image tokenizer is pivotal in both AR and MAR paradigms, as it significantly influences the quality of image generation. Despite the availability of various tokenizers (Esser et al., 2021; Chang et al., 2022; Sun et al., 2024; Luo et al., 2024) with reducing reconstruction errors, they often demonstrate inconsistent performance in downstream text-to-image generation tasks. This raises key questions: *Which tokenizer is the most effective for image generation tasks, and how can tokenizers be designed to be both efficient and practical?*

In this paper, we first conduct a comprehensive study on four prevalent tokenizers, including MaskGIT-VQ (Chang et al., 2022), Chameleon-VQ (Team, 2024), LlamaGen-VQ (Sun et al., 2024) and Open-MAGVIT2-VQ (Luo et al., 2024), across both AR and MAR generation paradigms. We observe that as the codebook size[2] increases (from 1,024 to 262,144), the tokenizer's image reconstruction ability improves progressively. However, this does not always lead to better image generation quality, as larger codebook sizes place higher demands on the model's learning ability and training parameters. Among the evaluated tokenizers, LlamaGen-VQ, with a codebook size of 16,384, demonstrates the best performance in both AR and MAR generation.

Building on these insights, we further explore the design space of MAR models to achieve efficient and scalable image generation. While previous MAR models have utilized the Bidirectional Transformer architecture, the success of LLaMA (Touvron et al., 2023) in AR generation inspires us to develop a Bidirectional LLaMA architecture. Through comparative analysis, we find that the Bidirectional LLaMA demonstrates superior image generation capabilities. To ensure stable training at larger scales, we introduce query-key normalization and post-normalization to manage the norm growth (Dehghani et al., 2023; Team, 2024; Gao et al., 2024). Based on this architecture, we present a series of class-conditional image generation models, ranging from 111M to 1.4B parameters. Additionally, we provide a text-conditional image generation model with 775M parameters, capable of flexibly generating images at various resolutions with high aesthetic quality.

Compared to AR generation, MAR generation has made a significant qualitative leap in speed. However, despite our comprehensive exploration of MAR's generation capabilities, it remains a performance gap compared to AR. To harness the strengths of both paradigms, we propose a hybrid framework during the inference stage. Specifically, we use an AR model to generate a portion of the tokens, which then serve as prompts for the MAR model to complete the remaining tokens. Our experiments on class-conditional and text-conditional generation tasks indicate that this strategy effectively strikes a balance between generation quality and speed, presenting a promising avenue for future research.

In summary, our contributions to the community include:

- **Discrete-Value Image Tokenizer Evaluation.** We evaluate the performance of four mainstream image tokenizers, with codebook sizes ranging from 1,024 to 262,144, across both AR and MAR generation paradigms. Through this evaluation, we identify the most ef-

---

[2]Codebook size design plays a critical role in determining image tokenization performance.

fective image tokenizer for image generation, providing insights into the optimal balance between codebook size and generation quality in both paradigms.

- **Scalable Mask Autoregressive Model Architecture.** We identify the optimal MAR model architecture and develop a series of class-conditional image generation models with parameter sizes ranging from 111M to 1.4B. The largest model achieves 3.71 FID on the ImageNet 256×256 benchmark, requiring only 8 inference steps. We further develop a text-to-image MAR model equipped with 775M parameters, unlocking efficient generation of photorealistic images at arbitrary resolution.

- **A Unified Framework for Fusion AR and MAR Generation.** To generate images both efficiently and with high quality, we propose a fusion framework that combines AR and MAR paradigms during the inference stage. By leveraging partial tokens generated by AR as initialization for MAR, we achieve a seamless transition between these two paradigms, which offers a flexible trade-off between efficiency and performance.

## 2 PRELIMINARIES

### 2.1 DISCRETE-VALUE IMAGE TOKENIZER

To begin with, we revisit the discrete-value image tokenizer, which plays a crucial role in both autoregressive and mask autoregressive image generation. The most commonly used model in this domain is the VQ-VAE (Van Den Oord et al., 2017), an encoder-quantizer-decoder architecture, as shown in Figure 2a. This architecture employs a ConvNet for both the encoder and the decoder, featuring a downsampling ratio $p$. The encoder projects the image pixels $x \in \mathbb{R}^{H \times W \times 3}$ to a feature map $f \in \mathbb{R}^{h \times w \times C}$, where $h = H/p$ and $w = W/p$. The core of this process lies in the quantizer, which includes a codebook $Z \in \mathbb{R}^{K \times C}$ with $K$ learnable vectors. Each vector $f^{(i,j)}$ in the feature map is mapped during quantization to the code index $q^{(i,j)}$ of its nearest vector $z^k$ in the codebook. Consequently, the image pixels $x \in \mathbb{R}^{H \times W \times 3}$ are efficiently quantized into $q \in \mathbb{Q}^{h \times w}$. During the decoding phase, the code index $q^{(i,j)}$ is remapped to the feature vector and the decoder converts these feature vectors back to the image pixels $\hat{x}$. There are various works (Razavi et al., 2019; Esser et al., 2021; Yu et al.) that continue to explore the design space of VQ-VAE for improving reconstruction quality. Among them, VQGAN (Esser et al., 2021) introduces an adversarial loss that is proven to effectively preserve the perpetual visual details.

### 2.2 AUTOREGRESSIVE GENERATIVE MODELS

Autoregressive models revolutionize the fields of language modeling (Brown et al., 2020; Achiam et al., 2023; Team et al., 2023; Touvron et al., 2023; Meta, 2024) and multimodal understanding (Liu et al., 2023; Lin et al., 2023; Team, 2024) using the unified ***next-token prediction paradigm for all modalities with a casual transoformer***, as illustrated in Figure 2b. This paradigm has been extended to the visual generation domain by early works such as DALL-E (Ramesh et al., 2021a), Cogview (Ding et al., 2021), and Parti (Yu et al., 2022). These works leverage a two-stage approach where the image tokenizer first encodes continuous images into discrete tokens then the transformer decoder models the flattened one-dimensional sequences. During training, the casual transformer is trained to predict the categorical distribution $p_\theta(x_i | x_{<i}; c)$ at each position conditioned on the additional information $c$, e.g., text prompts or class labels. During inference, image token sequences can be sampled in the same way as language generation and further decoded back to pixels. Despite this simple and unified paradigm for image synthesis, autoregressive-based visual generative models have long been overlooked for a while, particularly after the exploding of diffusion models. One potential reason is their inferior generation quality restricted by the image tokenizer. Recently, LlamaGen (Sun et al., 2024) improves the design of image tokenizer and leverages the Llama architecture for scaling, which significantly enhance the performance of autoregressive models.

### 2.3 MASK AUTOREGRESSIVE GENERATIVE MODELS

Different from the next token prediction paradigm typical of autoregressive generation, mask autoregressive models (Chang et al., 2022; Li et al., 2023; Chang et al., 2023) leverage a ***bidirectional***

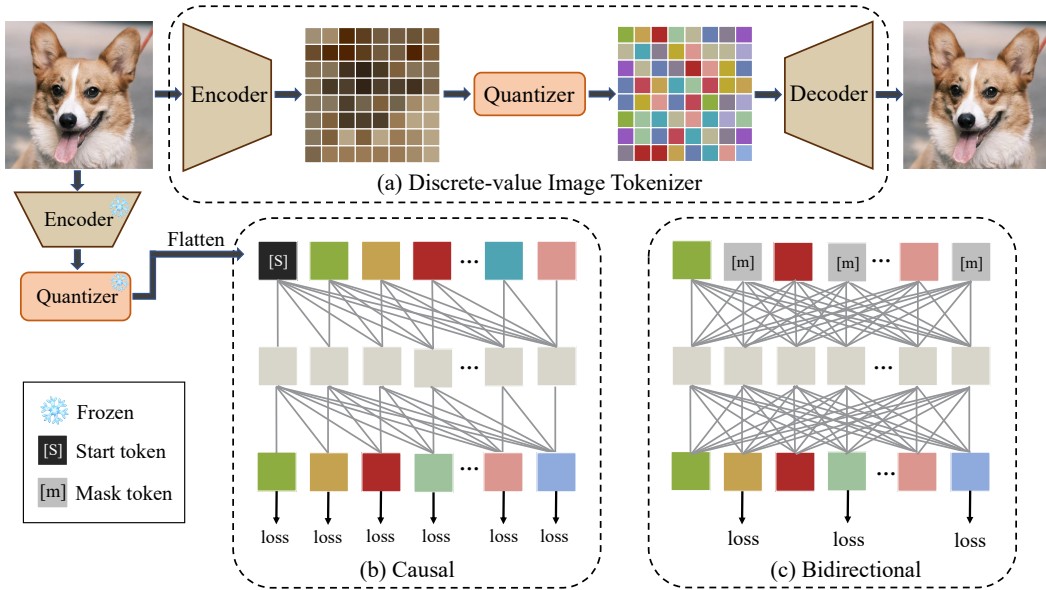

Figure 2: Illustration of (a) discrete-value image tokenizer (encoder and quantizer) and decoder via image reconstruction, (b) training the autoregressive model through causal attention modeling and (c) training the mask autoregressive model through bidirectional attention modeling.

***transformer to simultaneously generate all visual tokens through a masked-prediction mechanism***, as illustrated in Figure 2c. These models are trained using a proxy task similar to the mask prediction task employed in BERT (Kenton & Toutanova, 2019). In this setup, bidirectional attention allows all known tokens to see each other while also permitting all unknown tokens to view all known tokens, enhancing communication across tokens compared to causal attention. In contrast to causal attention, where training loss is computed on sequentially revealed tokens, non-autoregressive models compute the loss solely on the unknown tokens. At inference time, these models utilize a novel decoding method that synthesizes an image in a constant number of steps, typically between 8 and 15 (Chang et al., 2022). Specifically, during each iteration, the model predicts all tokens in parallel, retaining only those predicted with high confidence. Less certain tokens are masked and re-predicted in subsequent iterations. This process repeats, progressively reducing the mask ratio until all tokens are accurately generated through several refinement iterations.

## 3 RETHINKING IMAGE TOKENIZER FOR HIGH-QUALITY GENERATION

For high-quality image generation, the choice of a discrete-value image tokenizer is critical, as it determines the upper limit of the generation quality. Various tokenizers have been developed, and in this study, we evaluate four mainstream tokenizers based on their codebook sizes, which range from 1,024 to 262,144. This range allows for a comprehensive analysis of how codebook size affects image generation quality. The tokenizers we selected include MaskGIT-VQ (Besnier & Chen, 2023), Chameleon-VQ (Team, 2024), LlamaGen-VQ (Sun et al., 2024), and Open-MAGVIT2-VQ (Luo et al., 2024), as shown in Table 1. All tokenizers use a downsampling ratio of 16 and are trained on the ImageNet dataset (Deng et al., 2009). Given a set of compact discrete image tokens, two prominent frameworks are commonly used, including autoregressive and mask autoregressive generation. We conduct detailed experiments using both frameworks to provide comprehensive insights into image tokenizers.

### 3.1 VISUAL RECONSTRUCTION EVALUATION

Before assessing the impact of the image tokenizer on generation quality, we initially focus on evaluating the performance of each tokenizer itself. We analyze the reconstruction quality and codebook utilization using the ImageNet validation set, with results detailed in Table 1.

Table 1: **Reconstruction performance and codebook usage of different discrete-value image tokenizers on ImageNet validation set.** All tokenizers employ a downsampling ratio of 16 and are trained on the ImageNet training set at a training resolution of $256 \times 256$.

| Method | Tokens | Ratio | Train Resolution | Codebook Size | rFID↓ | Codebook Usage↑ |
|---|---|---|---|---|---|---|
| MaskGIT-VQ (Besnier & Chen, 2023) | $16 \times 16$ | 16 | $256 \times 256$ | 1024 | 10.79 | 44.3% |
| Chameleon-VQ (Team, 2024) | $16 \times 16$ | 16 | $256 \times 256$ | 8192 | 8.34 | 38.3% |
| LlamaGen-VQ (Sun et al., 2024) | $16 \times 16$ | 16 | $256 \times 256$ | 16384 | 4.54 | 100% |
| Open-MAGVIT2-VQ (Luo et al., 2024)] | $16 \times 16$ | 16 | $256 \times 256$ | 262144 | 4.03 | 100% |

From the results, we can find that as the codebook size increases, the relative rFID[3] value decreases, indicating that the reconstruction image quality is gradually improving. However, when the codebook size reaches a certain level, the improvement in the reconstructed image quality is limited. For instance, despite Open-MAGVIT2-VQ expanding its codebook size $16\times$ compared to LlamaGen-VQ, the rFID reduction is a mere 0.51, indicating that blindly increasing the codebook size for the sole purpose of improvement is not a wise choice in the design of a discrete-value image tokenizer. In terms of codebook utilization, LlamaGen-VQ and Open-MAGVIT2-VQ achieve 100%, which is primarily due to replacing traditional code assignment (*i.e.*, pair-wise distance) with lookup-free quantization (LFQ) (Luo et al., 2024).

## 3.2 VISUAL GENERATION EVALUATION

**Evaluation Setting.** To explore the impact of image tokenizers on image generation, we conduct detailed experiments using four tokenizers across two paradigms: Autoregressive (AR) and Mask Autoregressive (MAR) generation. For AR generation, we utilize the LLaMA as the foundational architecture, consistent with LlamaGen (Sun et al., 2024). For MAR generation, following MaskGIT (Chang et al., 2022), we employ the Bidirectional Transformer model as the foundational architecture and also modify LLaMA to a bidirectional variant. To facilitate a fair comparison between these foundational models, both the Transformer and LLaMA architectures are configured under identical conditions ($\sim$ 100M parameters): 12 layers, 12 heads, and 768 dimensions. All experiments are conducted on the class-conditional image generation ImageNet benchmark with $256\times256$ resolution and trained for 200 epochs. During the evaluation phase, we generate 50,000 images across 1,000 classes, with 50 images per class. We employ FID and Inception Score (IS) as evaluation metrics, consistent with previous studies (Li et al., 2024b).

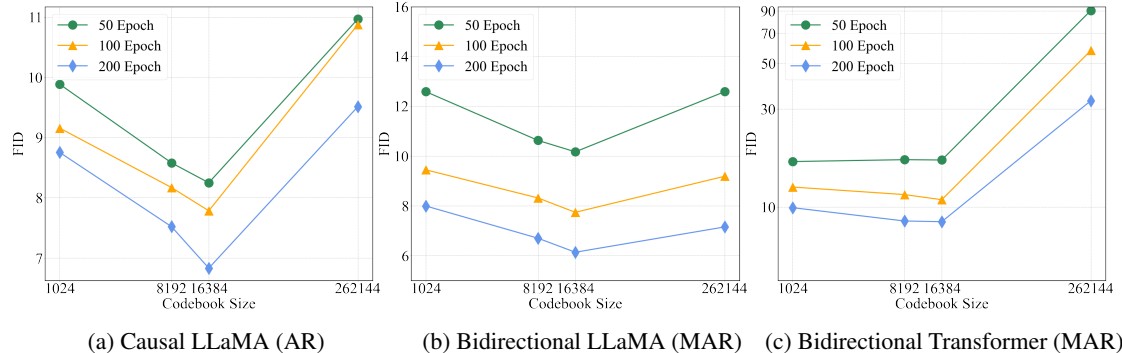

| (a) Causal LLaMA (AR) | (b) Bidirectional LLaMA (MAR) | (c) Bidirectional Transformer (MAR) |

Figure 3: **Visual Generation Evalutation.** We show FID of class-conditional ImageNet $256\times256$ benchmark over training epochs. More detailed experimental results are given in the Appendix.

**Evaluation Results.** The evaluation results are shown in Figure 3, and the detailed FID and IS results are shown in Appendix A. In our evaluation, we focus on answering two key questions:

---

[3]The reconstruction Fréchet inception distance, denoted as rFID (Heusel et al., 2017), is usually adopted to measure the quality of reconstructed images.

*Does increasing the codebook size improve generation quality?* In Section 3.1, we observe that image reconstruction quality is directly proportional to the codebook size, this is not the case for image generation quality. As shown in Figure 3, as the codebook size increases, the FID value initially decreases but then increases, both AR and MAR models exhibit this trend, which indicates that there is an optimal range for codebook size. The underlying issue is that when the number of learned or predicted codes becomes excessively large, it complicates model training. Therefore, when selecting an image tokenizer for generation tasks, it is crucial to consider the model's learning capacity in addition to the codebook size.

*Which tokenizer is the best for high-quality image generation?* Based on the results, LlamaGen-VQ emerges as the superior choice, delivering the best performance in both AR and MAR image generation tasks. Open-MAGVIT-VQ, which performs best in the reconstruction evaluation, delivers unsatisfactory results in image generation quality, especially on the Transformer architecture of the MAR generation task, where it performs very poorly. As for Chameleon-VQ, although it is widely adopted in many previous image generation works (Team, 2024; Liu et al., 2024), its performance does not match that of LlamaGen-VQ. Therefore, our exploration provides guidance for the selection of image tokenizers, suggesting that LlamaGen-VQ may be a better choice.

**Discussion.** Our evaluation focuses on generative models with ∼ 100M parameters, providing a valuable benchmark. While we do not rule out that scaling model parameters, MAGVIT2-VQ, with its 262,144 codebook size, could potentially yield better performance. However, its high training complexity, large parameters for the classification head, and significant GPU resource demands make it less practical for many applications. Further analysis is provided in the Appendix C.

# 4 SCALING MASK AUTOREGRESSIVE MODELS FOR IMAGE GENERATION

Numerous studies have extensively explored autoregressive generation, with notable contributions such as LlamaGen (Sun et al., 2024), which has successfully scaled autoregressive models to 3B parameters. However, the scaling of mask autoregressive models remains comparatively underexplored. To bridge this gap, we select a leading mask autoregressive model architecture and focus on scaling it. In this process, we employ QK-Norm and Post-Norm to maintain stability when scaling model to 1.4B parameters. Furthermore, we develop models for both class-conditional and text-conditional image generation.

## 4.1 IMAGE GENERATION BY MASK AUTOREGRESSIVE MODELS

**Autoregressive vs. Mask Autoregressive.** We compare the *Autoregressive (AR)* and *Mask Autoregressive (MAR)* generation paradigms under the same model architecture (LLaMA) and image tokenizers, as shown in Figure 4. Based on the FID scores, it is clear that the performance of the MAR paradigm is not inferior to AR, and in fact, it outperforms AR on certain image tokenizers. Additionally, MAR offers a substantial advantage in terms of inference steps, requiring only 8 inference steps, while AR requires 256. This significant reduction in inference steps underscores the speed and practicality of MAR in various applications. Despite these advantages, MAR models remain underexplored, and we aim to address this performance gap.

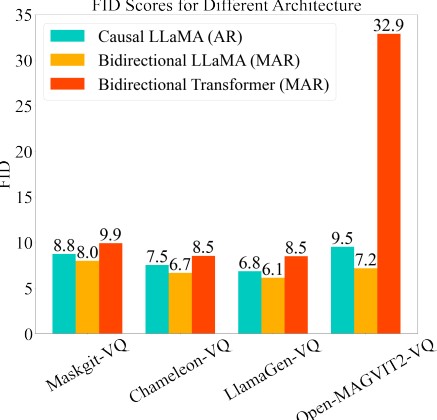

Figure 4: **The performance comparison between different architectures using four tokenizers.**

**Mask Autoregressive Model Architecture.** In Section 3.2, we explore the impact of image tokenizers on generation quality and confirm that LlamaGen-VQ is the most effective tokenizer. Furthermore, to determine the optimal MAR model architecture, we compare the *Bidirectional LLaMA* and *Bidirectional Transformer* architectures, as illustrated in Figure 4. The results clearly indicate that the Bidirectional LLaMA architecture outperforms the Bidirectional Transformer in terms of generation quality. Consequently, our model architecture is built on LlamaGen-VQ and Bidirectional LLaMA,

Table 2: **Model sizes and architecture configurations of our MaskGIL models.** The configurations are following previous works (Sun et al., 2024; Touvron et al., 2023; OpenLM-Research, 2023). We replace causal attention to bidirectional attention for mask autoregressive modeling. We add QK-Norm and Post-Norm for stable training.

| Model | Parameters | Layers | Hidden Size | Heads | QK-Norm | Post-Norm |
|---|---|---|---|---|---|---|
| MaskGIL-B | 111M | 12 | 768 | 12 | ✗ | ✗ |
| MaskGIL-L | 343M | 24 | 1024 | 16 | ✗ | ✗ |
| MaskGIL-XL | 775M | 36 | 1280 | 20 | ✗ | ✗ |
| MaskGIL-XXL | 1.4B | 48 | 1536 | 24 | ✔ | ✔ |

named **Mask**ed **G**enerative **I**mage **L**LaMA (**MaskGIL**). In LLaMA, we modify causal attention to bidirectional attention and incorporate 2D RoPE in every layer of the model, following the implementation in prior works (Lu et al., 2024; Fang et al., 2024). Additionally, we avoid using the AdaLN technique (Peebles & Xie, 2023) to maintain consistency with the standard LLM architecture.

**Classifier-Free Guidance.** Classifier-Free Guidance (CFG) (Ho & Salimans, 2021; Sanchez et al.) is originally proposed to improve the quality and text alignment of generated samples in text-to-image diffusion models. We integrate this technique into our MaskGIL models. During training, the conditional input is randomly dropped and replaced by a null unconditional embedding (Peebles & Xie, 2023; Gao et al., 2024; Liu et al., 2024; Zhuo et al., 2024). In practice, the probability of random drop is 0.1. During inference, for each image token, the CFG adjusts logits $\ell_{cfg}$, which are formulated as $\ell_{cfg} = \ell_u + s(\ell_c - \ell_u)$, where $\ell_c$ is conditional logits, $\ell_u$ is unconditional logits, and $s$ is the scale of the classifier-free guidance. As shown in Figure 5b, CFG has a significant impact on the generation performance of our MaskGIL models.

### 4.2 SCALE UP

**Scaling Mask Autoregressive Models.** In previous research on mask autoregressive image generation (Chang et al., 2022; Li et al., 2023), model sizes have typically ranged between 200M and 300M parameters, with limited exploration into scaling larger models. In this study, we investigate scaling the parameter size from 111M to 1.4B on the class-conditioned ImageNet generation benchmark. The detailed configurations of our MaskGIL models, with varying parameter sizes, are presented in Table 2. We also develop text-to-image MaskGIL models with 775M parameters.

**Training Stability.** Maintaining stable training while scaling MaskGIL models above 1.4B parameters proved challenging, with instabilities often emerging very late in the training process. This aligns with the observations in several prior studies (Dehghani et al., 2023; Team, 2024; Gao et al., 2024; Zhuo et al., 2024). The fundamental reason is that the standard LLaMA architecture for visual modeling shows complex divergences due to slow norm growth in the mid-to-late stages of training. We implement query-key normalization (QK-Norm) (Team, 2024; Dehghani et al., 2023) and Post-Norm (Zhuo et al., 2024) to solve this problem. QK-Norm involves applying layer normalization to the query and key vectors within the attention mechanism, while Post-Norm applies layer normalization to the output of attention and mlp layer, effectively avoiding the uncontrollable norm growth due to un-normalized pathway and therefore stabilizing the training process.

### 4.3 CLASS-CONDITIONAL IMAGE GENERATION

**Training and Evaluation Setup.** The class embedding is drawn from a set of learnable embeddings (Li et al., 2024a; Esser et al., 2021) and serves as the prefilling token embedding. From this initial token embedding, the model generates all image tokens. We conduct the class-conditional generation experiments on ImageNet dataset. All models are trained for token unmasking, using cross-entropy loss with label smoothing of 0.1. The optimizer employed is AdamW with learning rate of $1e^{-4}$, betas=(0.9,0.96) and a weight decay of $1e^{-5}$. For classifier-free guidance, we set the dropout rate for the class condition embedding to 0.1. All models are trained with a batch size of 256, which we identify as the optimal value through scaling experiments. We employ a weak-to-strong training strategy (Chen et al., 2024), gradually increasing the training resolution from $256 \times 256$

Table 3: **Model comparisons on class-conditional ImageNet 256×256 benchmark.** "Step" indicates the number of inference steps. Metrics include FID, IS, Precision and Recall. "↓" or "↑" indicate lower or higher values are better. More detailed results are in Appendix B.

| Type | Model | #Param. | Step↓ | FID↓ | IS↑ | Precision↑ | Recall↑ |
|------|-------|---------|-------|------|-----|-----------|---------|
| AR | RQTransfomer [Lee et al. (2022)] | 3.8B | 256 | 7.55 | 134.00 | – | – |
| | LlamaGen-B [Sun et al. (2024)] | 111M | 256 | 5.46 | 193.61 | 0.83 | 0.45 |
| | LlamaGen-L [Sun et al. (2024)] | 343M | 256 | 3.07 | 256.06 | 0.83 | 0.52 |
| | LlamaGen-XL [Sun et al. (2024)] | 775M | 256 | 2.62 | 244.08 | 0.80 | 0.57 |
| | LlamaGen-XXL [Sun et al. (2024)] | 1.4B | 256 | 2.34 | 253.90 | 0.80 | 0.59 |
| MAR | MaskGIT [Chang et al. (2022)] | 227M | 8 | 6.18 | 182.1 | 0.80 | 0.51 |
| | MAGE [Li et al. (2023)] | 230M | 20 | 6.93 | 195.8 | – | – |
| MAR | MaskGIL-B (CFG=2.0) | 111M | 8 | 5.64 | 229.96 | 0.83 | 0.48 |
| | MaskGIL-L (CFG=2.0) | 343M | 8 | 4.01 | 281.11 | 0.84 | 0.51 |
| | MaskGIL-XL (CFG=2.5) | 775M | 8 | 3.90 | 296.25 | 0.87 | 0.49 |
| | MaskGIL-XXL (CFG=2.5) | 1.4B | 8 | 3.71 | 303.47 | 0.88 | 0.52 |

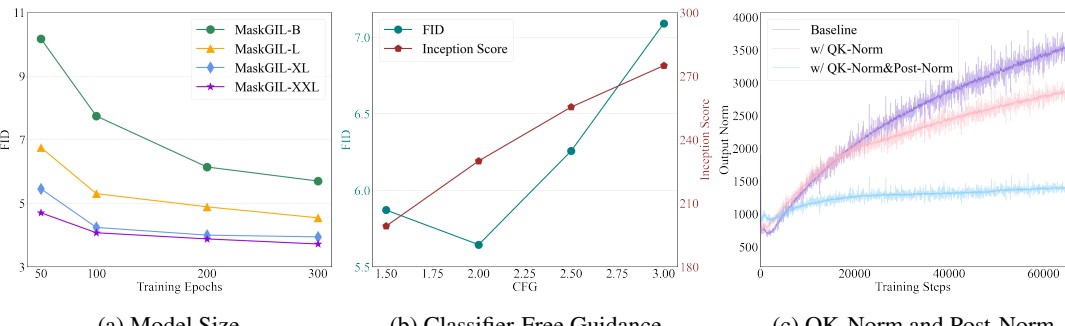

(a) Model Size        (b) Classifier-Free Guidance        (c) QK-Norm and Post-Norm

Figure 5: **The Effect of Model Size, CFG, QK-Norm, and Post-Norm.** We show the FID scores on the ImageNet benchmark across different model sizes and CFG configurations. Scaling the model size consistently improves FID scores throughout the training process. The impact of CFG is also notable. To monitor training stability, we plot the model's output norm.

to 512×512. In addition to evaluating performance using the FID and IS metrics, we also report Precision and Recall (Kynkäänniemi et al., 2019) to provide a more comprehensive assessment.

**Comparisons with Other Image Generation Methods.** In Table 3, we compare our model with popular AR and MAR image generation models, including RQTransformer (Lee et al., 2022), LlamaGen (Sun et al., 2024), MaskGIT (Chang et al., 2022), and MAGE (Li et al., 2023). Our MaskGIL models exhibit competitive performance across all metrics, including FID, IS, Precision, and Recall. Notably, MaskGIL requires only 8 inference steps, highlighting its efficiency in image generation. Further analysis of inference steps can be found in the Appendix D. A notable point is that the FID and Recall of MakGIL are slightly inferior to the LlamaGen, it demonstrates significantly high IS and Precision. This observation suggests a lack of diversity but high visual quality in generated samples, which can be fixed by exploring more advanced sampling strategies for MAR.

**Effective of Model Size.** We train our MaskGIL models across four model sizes (B-111M, L-343M, XL-775M, XXL-1.4B) and evaluate their performance in Table 3. Figure 5a illustrates how FID changes as both the model sizes and the training epochs increase. Notable improvements in FID are observed when scaling the model from MaskGIL-B to MaskGIL-XL. Further scaling to 1.4B yields only marginal improvements. A plausible explanation for this phenomenon could be the ImageNet data size limits the performance of the scaling model (Sun et al., 2024).

**Effect of Classifier-Free Guidance.** Figure 5b presents the FID and IS scores of MaskGIL-L under various classifier-free guidance (CFG) settings. MaskGIL-L achieves its best FID at CFG = 2.0, and increasing CFG beyond this point leads to a deterioration in FID, which is consistent with previous findings (Dhariwal & Nichol, 2021; Sun et al., 2024). This demonstrates that CFG plays a

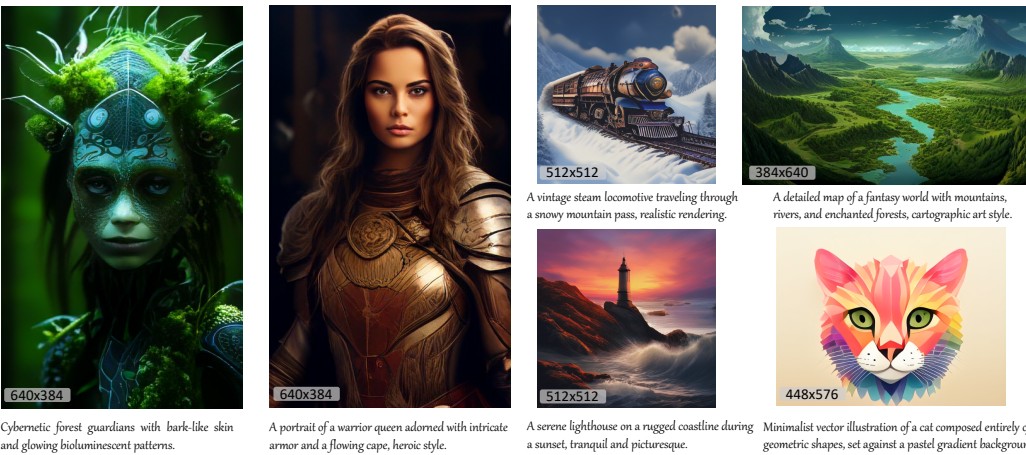

Figure 6: **Visualization of text-conditional image generation.** The prompts are generated by GPT-4. MaskGIL can generate images at any resolution while preserving consistency with the text.

crucial role in influencing the generation performance of our MaskGIL models. For further results on MaskGIL models of different sizes, please refer to the Appendix B.

**Effective of QK-Norm and Post-Norm.**    In Figure 5c, we show the output norm of the MaskGIL-XXL (1.4B) with and without QK-Norm/Post-Norm. Without QK-Norm and Post-Norm, the output norm grows uncontrollably, leading to non-convergence during training and the occurrence of "nan" loss values in the future training. Normalizing the query and key embeddings before computing the attention matrix helps to avoid attention collapse but merely mitigates this norm growth. Ultimately, the inclusion of both QK-Norm and Post-Norm ensures the output remains stable, preventing dramatic increases and resulting in more stable training for scaling.

### 4.4    Text-Conditional Image Generation

In text-conditional image generation, we use Chameleon-VQ as the image tokenizer, aligning with Lumina-mGPT, an autoregressive-based text-to-image model (Liu et al., 2024). To integrate the text condition into our MaskGIL model, we employ Gemma-2B (Team et al., 2024) as the text encoder. The encoded text features are projected through an additional MLP and then used as the prefilling token embedding in the MaskGIL models. We also incorporate both QK-Norm and Post-Norm to improve training stability. The training data consists of 20M high-aesthetic images with prompts generated by a mixture of captioners. We leverage a two-stage training pipeline by first training on $256 \times 256$ and then switching to $512 \times 512$. To enable generating images with arbitrary shapes, we design a multi-resolution training strategy by defining a series of size buckets and converting each image to its nearest shape. All other training hyper-parameters follow class-conditional settings.

In Figure 6, we use prompts randomly generated by GPT-4 to generate images at various resolutions. The generated images demonstrate a high degree of consistency with the text prompts. In the future, we plan to support higher resolution generation, such as $1024 \times 1024$, with better visual quality.

## 5    Unified Framework for Fusion of AR and MAR Generation

**Unified Framework at Inference Phase.**    In this work, we have thoroughly explored the generative capabilities of MAR and optimized them extensively. However, despite our advancements, a performance gap remains when compared to AR models. Drawing inspiration from recent work (Li et al., 2024b) that suggests AR and MAR can be unified into a single generative paradigm, we design a unified framework for the inference decoding stage. This framework effectively combines the strengths of both AR and MAR paradigms, achieving efficient generation with high quality. As illustrated in Figure 7, we initially employ AR to generate a subset of image tokens, which serve as prompts. Subsequently, MAR is deployed to complete the generation of the remaining tokens.

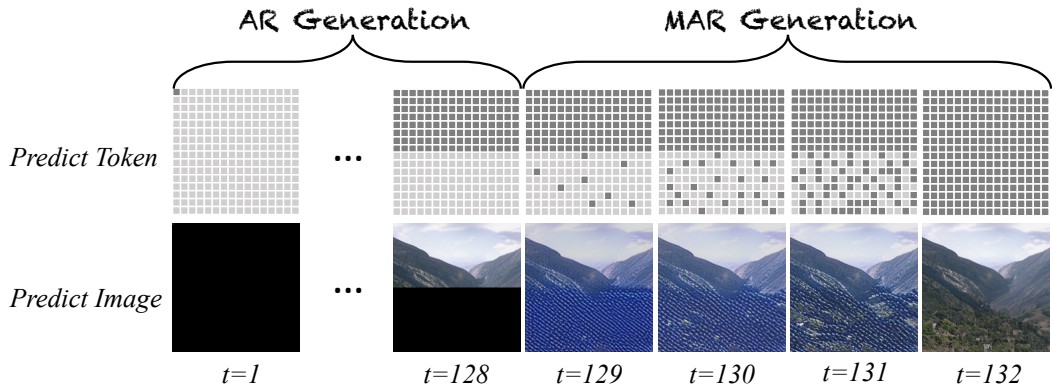

Figure 7: **llustration of our unified framework.** This framework initially employs AR to generate a subset of image tokens. Subsequently, MAR is deployed to complete the generation of the remaining tokens. In this example, only 132 inference steps are needed, whereas AR alone requires 256.

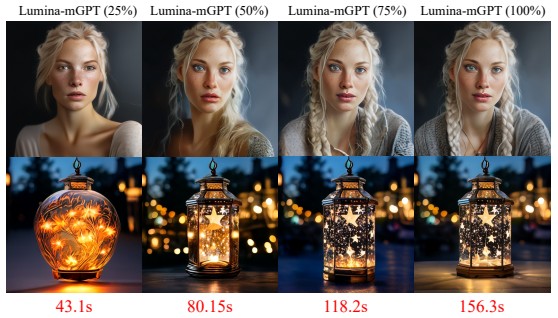

Figure 8: **Generated images and time at different AR ratios.** Our framework effectively reduces generation time while ensuring quality.

Figure 9: **Ablation study on the number of AR generated tokens.** Our framework achieves an optimal trade-off by generating 128 tokens with the AR.

**Application.** We apply this framework to both class-conditional and text-conditional image generation tasks. For the class-conditional AR model, we use LlamaGen (Sun et al., 2024), and for the text-conditional AR model, we employ Lumina-mGPT (Liu et al., 2024). In Figure 8, we showcase several examples along with the inference times for images generated using our framework. The results indicate a substantial improvement in inference speed while maintaining high-quality image generation. Additionally, we analyze the number of tokens generated by the AR component, as detailed in Figure 9. The convex function denotes an optimal point when generating 128 (50%) tokens with AR, offering a flexible trade-off between quality and efficiency.

# 6 CONCLUSION

In this work, we explore the potential of MAR models for efficient and scalable image generation. By reevaluating commonly used image tokenizers and model architectures, we identify the optimal MAR model configuration. We develop a variety of models for class-conditional image generation and introduce models capable of supporting text-conditional image generation. Our class-conditional models are competitive with popular AR models in terms of quality and demonstrate significant improvements in efficiency. Meanwhile, our text-conditional models maintain competitive visual quality and text alignment. Additionally, we explore the integration of AR and MAR paradigms at the inference stage, aiming to unify these approaches into a cohesive framework.

In the future, as more training data and computational resources become available, we will investigate large-scale MAR-based visual generation models, potentially exceeding 7B parameters. Furthermore, we intend to explore the end-to-end training that fuses AR and MAR paradigms.

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

## A    DETAILED EVALUATION RESULTS OF IMAGE TOKENIZER

### A.1    RESULTS ON AUTOREGRESSIVE GENERATION.

We present detailed experimental results of different image tokenizers on the AR paradigm in Table 4. We use the Casual LLaMA model for these experiments, evaluating on the class-conditional ImageNet benchmark. The results include FID and IS scores across 50, 100, and 200 epochs, with CFG settings of 3.0, 2.5, and 2.0. For a detailed analysis, refer to Section 3 in the main paper.

Table 4: **Detailed Experimental Results of Different Image Tokenizers on the AR Paradigm.** results include FID and IS scores for various CFG settings across 50 to 200 epochs.

| | | 50 epoch | | 100 epoch | | 200 epoch | |
|---|---|---|---|---|---|---|---|
| | | FID | IS | FID | IS | FID | IS |
| **MaskGIT-VQ (1024)** | CFG = 3.0 | 9.8846 | 160.1574 | 9.1567 | 170.2633 | 8.7525 | 175.5390 |
| | CFG = 2.5 | 10.2998 | 135.7815 | 9.4809 | 145.6140 | 9.1254 | 149.5679 |
| | CFG = 2.0 | 12.6750 | 102.6800 | 11.5306 | 110.0108 | 11.2182 | 114.8622 |
| **Chameleon-VQ (8192)** | CFG = 3.0 | 8.5799 | 180.8341 | 8.1700 | 193.9580 | 7.5208 | 200.0188 |
| | CFG = 2.5 | 8.9955 | 153.5894 | 8.2882 | 165.1865 | 7.7708 | 175.9126 |
| | CFG = 2.0 | 11.6736 | 115.8210 | 10.5847 | 125.7858 | 9.6992 | 133.8924 |
| **LlamaGen-VQ (16384)** | CFG = 3.0 | 9.2867 | 199.3415 | 8.8726 | 215.3014 | 8.0206 | 221.3254 |
| | CFG = 2.5 | 8.2499 | 171.4478 | 7.7843 | 185.0560 | 6.8314 | 191.7435 |
| | CFG = 2.0 | 8.7104 | 133.0047 | 7.8188 | 144.1255 | 7.0980 | 149.0620 |
| **Open-MAGVIT2-VQ (262144)** | CFG = 3.0 | 10.9688 | 129.7520 | 10.8770 | 131.0903 | 9.5125 | 146.1105 |
| | CFG = 2.5 | 15.5837 | 96.4929 | 14.8460 | 100.0897 | 14.8755 | 100.2676 |
| | CFG = 2.0 | 25.2163 | 63.6719 | 24.1729 | 66.1162 | 22.7751 | 71.5757 |

### A.2    RESULTS ON MASK AUTOREGRESSIVE GENERATION.

We present detailed experimental results of different image tokenizers on the MAR paradigm in Table 6. We use the Bidirection LLaMA and Bidirection Transformer model for these experiments, evaluating on the class-conditional ImageNet benchmark. The results include FID and IS scores across 50, 100, and 200 epochs, with CFG settings of 3.0, 2.5, and 2.0. For a detailed analysis, refer to Section 3 in the main paper.

## B    DETAILED EVALUATION RESULTS OF SCALING

In this work, we scale the model parameters of MaskGIT from 111M to 1.4B. Table 7 details the FID and IS scores for each model size at various training epochs and CFG settings. Notably, the results for our MaskGIT-XXL currently cover only up to 300 epochs. For a detailed analysis, please refer to Section 4 in the main paper.

## C    CODEBOOK SIZE AND TRAINING RESOURCE ANALYSIS

As detailed in Table 5, we compare the training resources required by LlamaGen-VQ and Open-MAGVIT2-VQ using the same model architecture (details in Section 3.2 of the main paper), both trained with a batch size of 192. From the results, it is evident that Open-MAGVIT2-VQ, with 201.32M parameters, has a parameter count 16 times greater than that of LlamaGen-VQ, which has 12.58M parameters. This substantial increase primarily stems from the final classification head. Additionally, Open-MAGVIT2-VQ requires twice the GPU resources compared to LlamaGen-VQ, indicating significant increases in both training parameters and memory usage as the codebook size expands.

Table 5: Codebook Size and Training Resource Analysis. We compare LlamaGen-VQ and Open-MAGVIT2-VQ.

| Method | LlamaGen-VQ | Open-MAGVIT2-VQ |
|---|---|---|
| Codebook Size | 16,384 | 262,144 |
| Classifier Params | 12.58M | 201.32M |
| A100 GPUs | 4 | 8 |

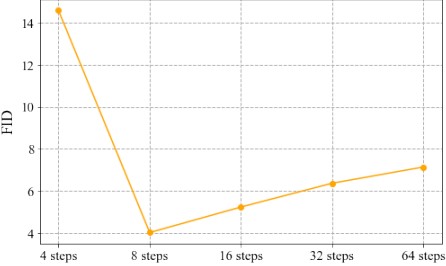

Figure 10: Ablation Study on Different Decoding Steps.

# D  DECODING STRATEGY AT INFERENCE PHASE

## D.1  DECODING STRATEGY

In autoregressive decoding, tokens are sequentially generated based on previously produced tokens. This method, inherently non-parallelizable, is particularly slow for images due to the typically large token length, such as 256 or 1024, which is much larger than that used in language. In this work, we adopt the decoding strategy outlined in Chang et al. (2022), where all image tokens are generated simultaneously in parallel. This is feasible due to the bi-directional attention of MaskGIL.

This decoding strategy allows for the generation of an image in T steps, typically 8. At each iteration, the model predicts all tokens simultaneously but retains only the most confident predictions. The less certain tokens are masked and re-predicted in subsequent iterations, with a progressively decreasing mask ratio, until all tokens are generated within the designated T iterations.

## D.2  ABLATION STUDY ON DECODING STEPS

In the main paper, all our experimental results are based on 8-step decoding. To further explore how the number of decoding steps affects generation quality, we conduct an ablation experiment on the class-conditional imagenet benchmark, as shown in Figure 10. The results indicate that increasing the number of decoding steps does not necessarily improve generation quality. Optimal results are achieved with 8-steps decoding.

Table 6: **Detailed Experimental Results of Different Image Tokenizers on the MAR Paradigm.** results include FID and IS scores for various CFG settings across 50 to 200 epochs.

| | | | MaskGIT-VQ (1024) | | | Chameleon-VQ (8192) | | | LlamaGen-VQ (16384) | | | Open-MAGVIT2-VQ (262144) | | |
|---|---|---|---|---|---|---|---|---|---|---|---|---|---|---|
| | | | CFG = 3.0 | CFG = 2.5 | CFG = 2.0 | CFG = 3.0 | CFG=2.5 | CFG=2.0 | CFG = 3.0 | CFG = 2.5 | CFG = 2.0 | CFG = 3.0 | CFG = 2.5 | CFG = 2.0 |
| **Bidirectional Transformer** | *50 epoch* | FID | 16.6349 | 17.9747 | 20.2007 | 16.9894 | 19.3724 | 22.9273 | 15.4235 | 16.9273 | 19.2510 | 90.4830 | 94.1686 | 98.6222 |
| | | IS | 98.2236 | 89.0695 | 76.8371 | 106.1228 | 92.8579 | 78.0924 | 114.4028 | 100.5291 | 86.2001 | 15.1394 | 14.1799 | 13.0694 |
| | *100 epoch* | FID | 12.4982 | 13.6014 | 15.5298 | 11.4843 | 12.7807 | 15.2883 | 10.7954 | 10.8447 | 11.6757 | 57.7810 | 64.0669 | 71.2787 |
| | | IS | 128.8805 | 114.9444 | 99.3552 | 148.9279 | 133.6862 | 112.5623 | 173.5138 | 156.9519 | 136.1681 | 31.4727 | 26.7601 | 22.9549 |
| | *200 epoch* | FID | 9.9150 | 10.8601 | 12.6045 | 8.5408 | 9.3071 | 11.2243 | 8.9188 | 8.4767 | 8.5895 | 32.8960 | 43.2282 | 51.0961 |
| | | IS | 154.5364 | 136.8136 | 115.7622 | 189.5333 | 169.3695 | 145.7171 | 213.7065 | 193.9830 | 169.5869 | 64.8568 | 45.3654 | 36.6035 |
| **Bidirectional LLaMA** | *50 epoch* | FID | 12.5828 | 13.3734 | 14.7296 | 10.6284 | 11.3143 | 12.9145 | 9.8001 | 9.7923 | 10.1693 | 12.5832 | 14.9028 | 18.8675 |
| | | IS | 121.7072 | 109.3891 | 96.8131 | 144.3818 | 130.8738 | 114.3169 | 169.3604 | 152.9462 | 136.0035 | 121.8998 | 104.8236 | 86.8462 |
| | *100 epoch* | FID | 9.4469 | 9.9894 | 10.9635 | 8.3184 | 8.4573 | 9.44030 | 8.1221 | 7.8057 | 7.7422 | 9.1883 | 11.2308 | 14.6781 |
| | | IS | 160.4776 | 143.2989 | 126.3950 | 188.8565 | 172.2684 | 152.5737 | 213.7423 | 194.7384 | 174.4664 | 153.0440 | 131.4824 | 108.0728 |
| | *200 epoch* | FID | 7.9954 | 8.3240 | 8.8536 | 6.7024 | 6.6765 | 7.2351 | 6.9971 | 6.4922 | 6.1397 | 7.1548 | 8.4162 | 9.3161 |
| | | IS | 191.4879 | 173.8534 | 154.6600 | 226.5536 | 206.9128 | 185.9040 | 244.6256 | 222.4675 | 201.6035 | 214.9400 | 172.6521 | 155.9095 |

Table 7: **Detailed Experimental Results of Scaling MaskGIL.** The results include FID and IS scores for various CFG settings across 50 to 400 epochs.

| | | MaskGIL-B (111M) | | | | MaskGIL-L (343M) | | | | MaskGIL-XL (775M) | | | | MaskGIL-XXL (1.4B) | | |
|---|---|---|---|---|---|---|---|---|---|---|---|---|---|---|---|---|
| | | CFG=3.0 | CFG=2.5 | CFG=2.0 | CFG=1.5 | CFG=3.0 | CFG=2.5 | CFG=2.0 | CFG=1.5 | CFG=3.0 | CFG=2.5 | CFG=2.0 | CFG=1.5 | CFG=3.0 | CFG=2.5 | CFG=2.0 |
| 50 epoch | FID | 9.8001 | 9.7923 | 10.1693 | 11.4326 | 8.1032 | 7.3123 | 6.7532 | 6.9289 | 5.9404 | 5.4523 | 5.4149 | 6.4069 | 4.9994 | 4.6997 | 5.0216 |
| | IS | 169.3604 | 152.9462 | 136.0035 | 115.6833 | 248.7714 | 229.2365 | 204.0812 | 172.9692 | 259.9695 | 235.1068 | 204.2972 | 172.1452 | 277.6597 | 251.9433 | 218.3809 |
| 100 epoch | FID | 8.1221 | 7.8057 | 7.7422 | 8.2393 | 6.8974 | 5.8711 | 5.3028 | 5.2657 | 4.7785 | 4.2401 | 4.2148 | 5.2318 | 3.8788 | 4.0727 | 4.8170 |
| | IS | 213.7423 | 194.7384 | 174.4664 | 150.8941 | 287.1859 | 265.8999 | 235.5590 | 203.3373 | 299.0290 | 271.2699 | 237.7044 | 198.0010 | 285.9541 | 255.0303 | 220.0994 |
| 200 epoch | FID | 7.4975 | 6.4922 | 6.1397 | 6.8546 | 6.2931 | 5.7252 | 4.8876 | 4.5048 | 4.2346 | 3.9977 | 4.1228 | 5.5923 | 3.9649 | 3.8788 | 4.4784 |
| | IS | 244.6256 | 222.4675 | 201.6035 | 169.0343 | 327.0787 | 303.9240 | 275.0859 | 235.0050 | 314.0859 | 281.0844 | 262.0203 | 205.5776 | 305.9541 | 274.0724 | 240.9620 |
| 300 epoch | FID | 6.9971 | 6.3837 | 5.0994 | 5.6713 | 6.1342 | 5.1874 | 4.5432 | 4.3718 | 4.1886 | 3.9442 | 4.0512 | 4.9251 | 3.8365 | 3.7199 | 4.2015 |
| | IS | 271.4360 | 250.5148 | 226.8228 | 194.3450 | 331.4310 | 306.8823 | 272.2563 | 236.6083 | 321.1712 | 291.0879 | 259.2443 | 220.4716 | 306.6679 | 282.4760 | 253.4782 |
| 400 epoch | FID | 7.0901 | 6.2576 | 5.6450 | 5.6713 | 5.4711 | 4.6372 | 4.0185 | 4.0854 | 4.0931 | 3.9031 | 4.0522 | 4.7214 | / | / | / |
| | IS | 274.9196 | 255.3847 | 229.9646 | 199.2374 | 336.8414 | 312.2809 | 281.1118 | 239.8479 | 326.2475 | 296.2475 | 269.8136 | 237.9921 | / | / | / |

