# OpenReview forum: "Exploring the Design Space of Autoregressive Models for Efficient and Scalable Image Generation"
_ICLR.cc/2025/Conference — ICLR 2025 Conference Withdrawn Submission_

### Official Review · Reviewer_b3sC · 2024-11-02

**Soundness:** 2
**Presentation:** 2
**Contribution:** 2
**Rating:** 3
**Confidence:** 4

**Summary:**

The paper "Exploring the Design Space of Autoregressive Models for Efficient and Scalable Image Generation" targets the investigation of a scalable setup for both autoregressive (AR) and masked autoregressive (MAR) models. To this end the paper starts out with an analysis of reconstruction accuracy of 4 Tokenizers across both paradigms, AR and MAR. Based on this the best model is scaled (by also including added normalization to the architecture), and performance to 4 methods from the literature analyzed. Finally, a unified AR/MAR geneartion framework is proposed.

**Strengths:**

+ the paper attempts to investigate different setups to scale for best quality in AR and MAR - a much needed investigation
+ good overview of existing methods, i.e. Figure 2 and corresponding text nicely elaborates on the existing context.
+ the paper is understandable very well

**Weaknesses:**

- The study on codebook size is confounded by the architectures of the tokenizers. To make a study on codebook size, a tokenizer must be selected, and then codebook size varied (all other things equal). If such an ablation study is not possible, an alternative would be to carefully rewrite the paper to make very clear that it analyzes from the set of 4 methods (and not the codebook size). E.g. statements like "when the codebook size reaches a certain level, the improvement in the reconstructed image quality is limited." cannot be claimed in generality from the given analysis. Please rephrase throughout the whole paper.
- Figure 4 is redundant to Figure 3. The 200 Epoch slice corresponds to shown bar plots. This is confusing to readers, in particular if skimming the paper figures. Please remove, or add it with explicit caption a subfigure to the same Figure.
- The proposed AR/MAR framework is too briefly analyzed. E.g.the proposed method suggests FID scores might be different in the AR part vs. the MAR part of the image. This should be analzyed. Also, if this turns out to be true, the AR part could be spaced out to cover the whole image (as 'seed' generation), which will likely improve quality. A plot that shows FID vs. generation time is missing (Figure 8 gives some intuition, but quantitative evidence is needed).
- Some formulations must be changed in the paper. Abstract: "we introduce modifications like query-key normalization and postnormalization" sounds like the paper invents these methods. Please make clear that this is not the case, e.g. "we adapt recently proposed methods like query-key normalization, ...". Also, please check some typos, e.g. "casual transoformer".
- Overall, it seems to me that too much is packed into this paper. E.g. the AR/MAR combination could be its own paper if pushed more, and investigated properly (some with other topics).

**Questions:**

- Generation. Unclear relation to MAGVIT-v2 (Yu et al - note that this reference lacks the year number in your references!): they showed that up to a vocabulary of 2^16=65k, generation performance improves monotonically. Are you claiming that this scaling behavior brakes down above 2^16? If so, there must be evidence - i.e. the plot from Magvit2 (Fig1 in their paper) should ideally be reproduced, or at least discussed in the paper.
- Why is section 4.4. proposing to use Chameleon-VQ - the paper before seems to lay out that LlamaGen-VQ is the better choice. Please discuss and justify in the paper, or make the corresponding experiments with LLamaGen-VQ.
- Figure 9: "Our framework achieves an optimal trade-off by generating 128 tokens with the AR.". Why is this the case? This remains elusive to the reader, as for an optimum one would expect a maximum/minimum in the plot (which is not at 128). This hence needs elaboration in the caption of the Figure.

---

### Official Review · Reviewer_pNsw · 2024-11-02

**Soundness:** 2
**Presentation:** 2
**Contribution:** 1
**Rating:** 3
**Confidence:** 4

**Summary:**

This paper explore the design space of image AR models. It first evaluate the codebook size of image tokenizers. Based on this, this work proposes MaskGIL, which uses MAR method to ensure few steps, and conduct study on its scaling property. The authors also propose a sampling framework that combines AR and MAR sampling.

**Strengths:**

1. The paper investigate the relationship between the performance and codebook size.
2. QK-Norm and Post-Norm methods are used to enhance the training stability.
3. It conduct experiments on text-to-image task and support generation at arbitrary resolutions.

**Weaknesses:**

1. The main concern is about the limited novelty and contribution. Most of the contents in this paper have already been studied
in previous works, as discussed below:
    - **Study of codebook size**. A similar study has already been conducted in Sec 3.1 of LlamaGen [1]. It seems that the biggest difference in this paper’s study is the attempt to use an ultra-large codebook size and show that it doesn't work. However, this conclusion is not surprising since such a large size makes it challenging for the model to learn effectively.
    - **Combining MAR (and the bidirectional architecture) with LlamaGen**. The authors mention in line 315 that the MAR method remains unexplored, but as far as I know, there has already been systematic research [2] on this, which is not discussed by the authors. Compared to [2], this paper’s method seems to simply replace its novel diffusion loss with the traditional cross-entropy loss. However, the usage of discrete tokens and cross-entropy loss is what [2] explicitly criticizes and tries to address.
    - **Effect of Scaling Up and CFG**. Popular previous works like MAR [2], VAR [3], and LlamaGen [1] all provide results for scaled-up models (e.g., Table 4 in [2], Table 6 in [1], Table 1 in [3]). Additionally, they all fully explore the effects of CFG (Tables 1, 2, and 3 in [2]; Table 3 in [3]; and Fig. 3(a) in [1]). Therefore, improvements gained by increasing the parameter size or using CFG are expected. This paper fails to present new knowledge or insights.
    - **Unified sampling framework**. Since MAR naturally supports predicting any number of tokens [2,4], adjusting the number of tokens generated per step is straightforward. [2] has already explored a cosine schedule for the number of tokens generated at each step. However, in this paper, the authors seems to simply adopt a basic approach of generating 1 token per step first and then turn to multiple tokens per step, without discussing the advantages of this method or comparing it to previously explored schedules.
Given these reasons, I recommend that the authors reconsider their claims of novelty.
2. Some statements need further clarification:
    - What differences do the tokenizers studied in this paper have besides codebook size? If there are other significant differences, it is better for the authors to list them and the comparison on codebook size alone might not be valid.
    - How is MaskGIL trained? Does it follow the traditional MAR approach of masking a part of the image each time and predicting this part based on the rest part? The training process does not seem to be mentioned in the paper (please correct me if I missed it).

[1] Sun et al., Autoregressive Model Beats Diffusion: Llama for Scalable Image Generation

[2] Li et al., Autoregressive Image Generation without Vector Quantization

[3] Tian et al., Visual Autoregressive Modeling: Scalable Image Generation via Next-Scale Prediction

[4] Chang et al., MaskGIT: Masked Generative Image Transformer

**Questions:**

Please refer to the Weakness.

---

### Official Review · Reviewer_oCKp · 2024-11-03

**Soundness:** 2
**Presentation:** 3
**Contribution:** 2
**Rating:** 5
**Confidence:** 4

**Summary:**

This paper explores a comprehensive recipe for AR models, including the selection of image tokenizers, model architectures, and AR paradigms. This paper conducts a detailed evaluation of four image tokenizers in both AR and MAR settings, investigating the impact of codebook size (varying from 1,024 to 262,144) on generation quality and identifying the most effective tokenizer for image generation. Subsequently, they propose an enhanced MAR model architecture, dubbed MaskGIL. Experiments on ImageNet demonstrate the effectiveness and superiority of this approach.

**Strengths:**

1. This paper investigates the impact of codebook size on generation quality within discrete image tokenizers and determines the optimal size for the codebook.
2. This paper proposes the application of QK-Norm and Post-Norm to achieve stable training of a large-scale 1.4B model, facilitating the scaling up of the model.
3. The writing style of this paper is clear, making it easy to understand.

**Weaknesses:**

1. The innovation is limited. Although the exploration of discrete Image tokenizers is meaningful to me, the rest of the content, whether it's about scaling up or the Unified Framework at the Inference Phase, seems to be a direct concatenation of existing analyses and methods. I did not observe more meaningful or deeper insights. For example: (1) The combination of QK-Norm and Post-Norm mentioned in the text can make the model training more stable. In fact, QK-Norm and Post-Norm are often used in the training of large models. Therefore, I would like to see more analysis on "the standard LLaMA architecture for visual modeling shows complex divergences due to slow norm growth in the mid-to-late stages of training." Whether this is only the case with the LLaMA framework, or whether there are other solutions, or whether this complex divergence is related to data? (2) Why the simple priority of AR before MAR can improve the quality of generation? Does MAR require a more accurate prior? What is the impact of the cumulative error of AR inference?
2. The comparative experiments for LLMs are limited, involving only Llama and Transformer. The recently strong QWen series and internLM series have not been included in the comparison. The motivation of this paper is to find the optimal combination of MLLMs, which means that it is necessary to compare the combination effects of different LLMs as much as possible. The current QWen2 and IntenrLM2 are both relatively advanced open-source models, which are worth comparing and analyzing.
3. The evaluation method is overly simplistic, with assessments conducted solely on ImageNet. It would be beneficial to see metrics such as FID on datasets like MSCOCO 30K. I would like the authors to explain why you have chosen to focus on ImageNet, and whether there are any other constraints that were considered?

**Questions:**

## Justification For Recommendation And Suggestions For Rebuttal：
- Justification For Recommendation：Reference to Paper Strengths.
- Suggestions For Rebuttal:
  1. The analysis of the Impact of different model architectures needs to be provided.
  2. The analysis of experimental results needs to be more detailed.
  3. Present more experimental data to support the motivations behind the paper.

## Additional Comments For Authors：
To enhance clarity and persuasiveness, the authors should rectify vague descriptions and inaccuracies in the details. For example, at line 278, "Based on the results" should specify which table or figure is being referred to. It is recommended that the authors review the entire paper to identify similar instances and make corrections.

---

### Official Review · Reviewer_eYQF · 2024-11-06

**Soundness:** 3
**Presentation:** 3
**Contribution:** 2
**Rating:** 5
**Confidence:** 4

**Summary:**

In this work, the authors investigate the design space of autoregressive image generation (i.e., AR and MAR). On ImageNet experiments, the authors report that larger codebook size doesn't always lead to better performance. It applies bidirectional LlaMA for masked AR and achieves better performance than using vanilla Transformer. Also, the authors propose to combine MAR and AR in inference to improve the performance of MAR.

**Strengths:**

1. The work investigates a valuable problem: how to design efficient and powerful autoregressive image generation model.
2. The work includes benchmark on standard ImageNet 256.
3. The work includes experiments with model at different scales from ~100M to 1.4B.

**Weaknesses:**

The major concern is that though the work tries to investigate a valuable problem, the analysis is not comprehensive enough. From my perspective, I find it a bit overclaimed by entitling as "exploring the design space of autoregressive models". For example, the paper demonstrates better performance of LlaMA over vanilla Transformer in MAR modeling. However, there are multiple modifications in LlaMA compared to Transformer yet the authors haven't further explore to figure out what is the key factors in design space. For instance, LlaMA applies rotatory positional embedding and SwiGLU FFN instead of absolute positional embedding and ReLU FFN. It would be valuable to further isolate the effect of key components in LlaMA.

**Questions:**

1. In Table 1, there are other factors besides codebook size of different tokenizers, e.g., LFQ. How does these affect the performance of reconstruction and generation? In particular, the authors may conduct ablation study of LFQ with codebook size 8192 to see how it compares to Chameleon-VQ.
2. In Table 3, there are other autoregressive models like VAR that are not included. Since VAR represents another design choice in image tokenizer, it would help understand the performance of proposed method. The authors may consider adding these results to the table.
3. For unified framework in inference phase, does it need two separate AR and MAR? If that's the case, it significantly increases the computational overhead in training. The authors are suggested to report more comprehensive computational cost of AR/MAR models in both training and inference.
4. Different tokenizers are evaluated on ImageNet. How does scaling to large dataset affect the conclusion? Will that benefit from larger codebook? It would be valuable to compare tokenizers on large scaled dataset. If limited by computational resources, having results on smaller dataset (e.g., subset of ImageNet) and inferring the performance at larger scale would also be beneficial.

Reference:

[1] https://arxiv.org/abs/2404.02905

---

### Note · Authors · 2024-11-13

I have read and agree with the venue's withdrawal policy on behalf of myself and my co-authors.